# Mesenchymal Stem Cells Profile in Adult Atopic Dermatitis and Effect of IL4-IL13 Inflammatory Pathway Inhibition In Vivo: Prospective Case-Control Study

**DOI:** 10.3390/jcm11164759

**Published:** 2022-08-15

**Authors:** Anna Campanati, Monia Orciani, Andrea Marani, Mariangela Di Vincenzo, Simona Magi, Stamatios Gregoriou, Federico Diotallevi, Emanuela Martina, Giulia Radi, Annamaria Offidani

**Affiliations:** 1Dermatological Clinic, Department of Clinical and Molecular Sciences, Università Politecnica delle Marche, 60020 Ancona, Italy; 2Histology, Department of Clinical and Molecular Sciences, Università Politecnica delle Marche, 60126 Ancona, Italy; 3Pharmacology, Department of Biomedical Sciences and Public Health, Università Politecnica delle Marche, 60126 Ancona, Italy; 4Faculty of Medicine, 1st Department of Dermatology-Venereology at Andreas Sygros Hospital, National and Kapodistrian University in Athens, 16121 Athens, Greece

**Keywords:** psoriasis, atopic dermatitis, mesenchymal stem cells, dupilumab, biologics, therapy, regenerative medicine

## Abstract

Atopic dermatitis (AD) is an inflammatory disease that typically begins in childhood and may persist into adulthood, becoming a lifelong condition. The major inflammatory mediators of AD are known to be interleukin IL4 and IL13, so Dupilumab, which is able to inhibit both interleukins by blocking the shared IL4Rα subunit, has become an attractive option for treating AD. Mesenchymal stem cells (MSCs) are involved in the onset and development of AD by secreting specific interleukins. The aim of this study was to isolate MSCs from healthy controls (C-MSCs) and patients with AD before (AD-MSCs T0) and after 16 weeks of treatment with Dupilumab (AD-MSCs T16); to evaluate the expression mainly of IL4 and IL13 and of other inflammatory cytokines in C-MSCs, AD-MSCs at T0 and at T16; and to evaluate the efficacy of Dupilumab on MSCs immunobiology. C- and AD-MSCs (T0, T16) were isolated from skin specimens and characterized; the expression/secretion of IL4 and IL13 was evaluated using immuno-cytochemistry (ICC), indirect immune-fluorescence (IIF) and an ELISA test; secretion of IL2, IL4, IL5, IL6, IL10, IL12, IL13, IL17A, Interferon gamma (IFNγ), Tumor necrosis factor alpha (TNFα), Granulocyte Colony-Stimulating Factor (G-CSF), and Transforming Growth Factor beta1 (TGFβ1) were measured with ELISA. IL13 and IL6 were over-expressed, while IL4 was down-regulated in AD-MSCs at T0 compared to C-MSCs. IL6 and IL13 expression was restored after 16 weeks of Dupilumab treatment, while no significant effects on IL4 expression were noted. Finally, IL2, IL5, IL10, IL12, IL17A, INFγ, TNFα, G-CSF, and TGFβ1 were similarly secreted by C- and AD-MSCs. Although Dupilumab blocks the IL4Rα subunit shared by IL4 and IL13, it is evident that its real target is IL13, and its ability to target IL13 in MSCs reinforces the evidence, already known in differentiated cells, of the central role IL13 rather than IL4 in the development of AD. The inflammatory cascade in AD begins at the mesenchymal level, so an upstream therapeutic intervention, able to modify the immunobiology of atopic MSCs, could potentially change the natural history of the disease.

## 1. Introduction

Atopic dermatitis (AD) is a systemic and immune-allergic inflammatory skin disease; it usually appears in early childhood (15% to 30%) and generally resolves before puberty. However, in more than half of patients, it can persist into adulthood, becoming a permanent condition [1,2].

Although prevalence of adult AD remains unclear, several studies have indicated that it has increased in recent decades, particularly in industrialized countries [3]. Current estimates place prevalence of AD at around 2–8% in adults, compared to 10–20% in children [4,5].

In AD, the breakdown of the skin barrier results in increased trans-epidermal water loss, reduced skin hydration, and improved antigen presentation by Langerhans cells, which initiate inflammation [6,7,8].

The largely activated mechanism is the T helper type 2 (Th2) and T helper type 22 (Th22) cascade, with consequent release of cytokines (e.g., interleukin (IL)4, IL13, IL2, IL10, IL17, IL22, Interferon gamma (IFNγ), Tumor necrosis factor alpha (TNFα), Granulocyte Colony-Stimulating Factor (G-CSF), and Transforming Growth Factor beta1 (TGFβ1) [9,10]. In active AD, Th2 inflammation and barrier disruption are characterized by reduced filaggrin and claudin 1 expression, resulting in further exacerbation of the barrier defect and enhanced risk of development of asthma and hay fever, as well as transcutaneous sensitization to a variety of food allergens (e.g., peanuts) [9,11,12].

Although all these immunologic features are well established in AD, the pathogenic model has profoundly changed in the last two decades, overcoming previous hypotheses based on the immune response mediated by immunoglobulin E (IgE) (type 1 hypersensitivity), the primary role of the epidermal barrier impairment (“outside-in” theory), and the primary role of the aberrant immune activation (“inside-out” theory). Previously [13,14], we demonstrated that skin-derived mesenchymal stem cells (MSCs)from AD patients showed strong differences compared to MSCs isolated from skin of healthy subjects. This finding suggests that MSCs may be involved in the pathogenesis of AD.

Dupilumab has entered the therapeutic armamentarium of AD in recent years, due to its selective action on IL4-IL13. Dupilumab is an IL4- receptor α-antagonist that inhibits IL4 and IL13 signaling by blocking the shared IL4 receptor α subunit. The blockade of IL4/13 is effective in reducing Th2 response.

In this study, the relative expression of selected Th1, Th2, and Th17 chemokines/cytokines has been analyzed in MSCs obtained from healthy subjects and adult AD patients, both before and after 16 weeks of treatment with Dupilumab. Recent studies on topical dermatitis, performed, for example, on skin biopsies or peripheral blood, have shown that the Th2 and Th17 responses are characteristic of the acute phase of disease, while the Treg and Th1 response are over-expressed in chronic phase of disease. Thus, it was our intention to evaluate the expression of these molecules in MSCs from patients with atopic dermatitis and compare it with that of MSCs from healthy patients, to delineate the immunophenotype of atopic stem cells in the different stages of disease [15].

## 2. Materials and Methods

### 2.1. Design of the Study

The study is a prospective case–control analysis approved by Polytechnic Marche University Ethical Commitee (Protocol 2016 0360 OR) and conducted according to the Declaration of Helsinki.

### 2.2. Patients’ Population

The study group included 11 adult patients (6 males, 5 females, mean age 46.7 ± 12.4), suffering from chronic AD (mean duration of disease 27.3 ± 13.2 years), whereas the control group consisted of 11 adult Caucasian healthy subjects (6 males, 5 females, mean age 49.5 ± 11.7). Diagnosis of AD in adults was made by two independent trained dermatologists, according to Italian AD guidelines [16], and it was essentially based on the typical clinical signs and symptoms of the disease, as currently, no diagnostic markers are available. These include lichenified and minimally inflammatory features of the eczematous skin lesions, the lesions’ sharp margins and symmetrical distribution, prevalent localization on the backs of the hands and fingers and on the volar side of the wrists, the presence of severe pruritus, a chronic clinical course with temporary remissions during summer, association with atopic mucous manifestations, and a positive family history of atopy.

In all enrolled patients, the coexistence of allergic contact dermatitis has been excluded through standardized epicutaneous patch tests conducted according to the SIDAPA (Società Italiana di Dermatologia Allergologia Professionale e Ambientale) guidelines.

All atopic patients were asked to avoid sun exposure and the use of topical and systemic specific treatments (corticosteroids, antihistamines, UVA, PUVA, nb-UVB, cyclosporine, pimecrolimus, and tacrolimus) for at least 4 weeks.

The severity of adult AD for each subject was estimated according to the Eczema Area and Severity Index (EASI), SCORing AD (SCORAD), and Investigator Global Assessment (IGA).

EASI is a validated investigator-assessed scoring system that, by grading the physical signs of atopic dermatitis, determines the severity of the patient’s eczema, according to a clinician’s perspective [17]; its final score rages from 0 to 72.

SCORAD index is a mixed patient/clinicians’ tool used to evaluate AD severity [18], and its final score ranges from 0 to 103.

The Physician Global Assessment (PGA), also referred to as IGA in clinical trials, assesses overall disease severity at a given timepoint on a 6-point severity scale, from clear to very severe disease [19]. Clinical characteristics of erythema, infiltration, papulation, oozing, and crusting are used as guidelines for the overall severity assessment.

### 2.3. Skin Samples

Atopic patients received a skin punch biopsy in lesional skin both before (T0) and after treatment with Dupilumab 600 mg (two 300 mg subcutaneous injections) once, and then 300 mg subcutaneous injection every other week for 16 weeks (T16).

Control healthy subjects, undergoing surgery for epidermal cysts, received one skin biopsy on healthy skin, after written informed consent was obtained. All punch biopsies were performed with a 5 mm sterile cutaneous skin punch biopsy device (Gima, medical devices, s.r.l. Rome, Italy) after local anesthesia with lidocain 2%. All specimens were obtained from the skin of the chest to standardize the skin specimens between patients and controls.

### 2.4. Isolation, Cell Culture and Characterization of MSCs

MSCs derived from skin samples were isolated, cultured as previously described [13,14,20], and characterized according to the criteria by Dominici [21] for the identification of MSCs. Briefly, after mincing, the samples were cultured with the Mesenchymal Stem Cell Growth Medium bullet kit (Euroclone, Milan, Italy). The morphology was assessed using phase contrast microscopy (Leica DM IL; Leica Microsystems GmbH, Wetzlar, Germany); the expression of cellular markers HLA-DR, CD14, CD19, CD34, CD45, CD73, CD90, and CD105 was measured using cytofluorimetric analysis to evaluate the immunophenotype, and osteogenic, chondrogenic, and adipogenic differentiation assays were performed as previously described [14]. Cells isolated from control healthy subjects and from AD patients were named C-MSCs and AD-MSCs, respectively.

### 2.5. IL4 and IL13 Expression by ICC and IIF

Considering the action of Dupilumab on IL4 and IL13, immunocytochemical and immunofluorescence analyzes for IL4 and IL13 were performed on C-MSC and AD-MSC at T0 and T16.

For Immuno-cytochemistry (ICC), 1.5 × 10^4^ cells were incubated overnight with anti-IL4 (R & D Systems, Minneapolis, Canada) or anti-IL13 (Santa Cruz Biotechnology, Dallas, TX, USA) primary antibody. Then, cells were immune-stained using the streptavidin–biotin–peroxidase technique (Dako Cytomation, Milano, Italy) and incubated with 3,3-diaminobenzidine. Slides were counterstained with Mayer’s hematoxylin.

For Indirect Immuno-Fluorescence (IIF), we incubated the same number of cells with the anti-IL4 or anti-IL13 antibody, followed by goat anti-mouse FITC-conjugated antibody. Nuclei were visualized using Hoechst 33342 (all from Sigma-Aldrich, St. Louis, MO, USA).

To quantify the expression of the proteins, the percentage of area occupied by the protein within the cells was calculated using Fiji-ImageJ software [22].

### 2.6. ELISA Test for Evaluation of Cytokines Levels in Supernatant

The levels of cytokines (IL2, IL4, IL5, IL6, IL10, IL12, IL13, IL17A, INFγ, TNFα, G-CSF, and TGFβ1) in the supernatant were determined using a commercial ELISA kit (Multi-Analyte ELISArray™ Kits, Qiagen Multi-Analyte ELISArray kit, Qiagen; ThermoFisher, Waltham, MA USA; Affymetric Ebioscences, Vienna, Switzerland) according to the manufacturer’s instructions. Next, 2.5 × 10^5^ cells were plated and cultured with 3 mL of medium; after 72 h, 50 μL of the supernatant was used for the ELISA test.

Briefly, samples (six C-MSCs and nine AD-MSCs at T0 and T16) were dispensed into a 96-well microtiter plate and incubated at room temperature for 2 h. Plates were then washed and reacted with avidin-HRP-conjugated antibody at room temperature for 30′.

After washing, captured cytokines were detected by addition of substrate solution. The OD at 450 and at 570 nm was determined using a microtiter plate reader (Multiskango microplate reader, Thermo Scientific). The concentration of cytokines was determined in pg/mL by comparing the absorbances with those of the antigen standards [23].

### 2.7. Statistical Analysis

All data were analyzed using Graph-Pad Prism (version 7.0, El Camino REAL, San Diego, CA, USA) and QuickCalcs software package. All data were expressed as means ± SD.

The distribution of continuous variables was verified with a Kolmogorov–Smirnov test. Since data did not assume Gaussian distribution, a nonparametric Kruskal–Wallis test was used for unpaired variables.

For correlation between variables, a computed nonparametric Spearman correlation was used. For all the analyses, a p-value less than 0.05 was statistically significant.

## 3. Results

### 3.1. Clinical Features

Treated patients were good responders to Dupilumab and achieved EASI 75 at T16 (Table 1).

All median values for clinometrics showed a significant improvement from baseline values at T16 (Table 2).

### 3.2. Isolation, Culture and Characterization of MSCs

MSCs were isolated from the skin of AD patients and control healthy subjects (Figure 1).

Isolated cells met the criteria by Dominici for MSCs identification: they were plastic adherent with a fibroblastoid morphology (Figure 1A), able to differentiate into osteoblasts (Figure 1B), chondrocytes (Figure 1C) and adipocytes (Figure 1D), and with a stem-like immunophenotype (Figure 1E). No differences were found among C-MSCs, AD-MSCs T0, and AD-MSCs T16.

### 3.3. IL4 and IL13 Expression by ICC and IF in C-MSCs and AD-MSCs at T0 and T16

Since Dupilumab specifically modulates IL4 and IL13, as demonstrated by several previous studies [24,25] their expression was evaluated by ICC and IIF in C-MSC and AD-MSC before and after drug treatment, followed by quantification using Fiji-ImageJ software [22]. Both experimental approaches revealed a significant increase in IL13 expression in AD-MSC (T0) compared to control. After 16 weeks of drug treatment, IL13 expression was restored to levels closer to controls (Figure 2).

In contrast, IL4 expression was significantly reduced in AD-MSC (T0) compared to controls, and drug treatment did not produce any significant change in its expression (Figure 3).

### 3.4. Expression Profiles of Th1, Th2, Th17 Cytokines by ELISA in C-MSCs and AD-MSCs at T0 and T16

The secretion of 12 cytokines (IL2, IL4, IL5, IL6, IL10, IL12, IL13, IL17A, INFg, TNFa, G-CSF, TGFβ1) was measured by ELISA test in the supernatant of 6 samples of C-MSCs and 9 samples of AD-MSCs, both at baseline and after 16 weeks of treatment with Dupilumab.

IL6 and IL13 were more secreted by AD-MSCs than C-MSCs at T0, and their secretion was directly correlated with disease severity at baseline, according to clinometric indexes EASI, SCORAD and IGA (except for IL6 and IGA) (Figure 4 and Figure 5). The secretion of IL6 and IL13 was significantly reduced after 16 weeks of treatment with Dupilumab, reaching levels closer to controls (Figure 4 and Figure 5; Table 3).

## 4. Discussion

Adult AD is a systemic, immune-allergic inflammatory skin disease; the inflammatory pathway most involved in development of AD is the type (Th) 2 immune response, although it has recently been shown that type 1 immune cells contribute to the chronic phase of AD [25].

The course of AD is characterized by biphasic inflammation, the Th2 and Th22 inflammatory pathways predominate at the onset and in the acute phases of disease, with an increase in tissue and serum levels of several cytokines, including IL4, IL5, IL13, IL22, IL31, and TSLP [26,27]. In chronic skin lesions, the prevalence of Th1/Th0 pathways has been described with increased production of IFNγ, TNFa, IL6, IL12, IL17A, and G-CSF [28,29,30,31].

The reservoir of MSCs in the skin has been extensively studied in order to understand their possible role in the pathogenesis of several skin diseases, since MSCs are known to modulate the innate and adaptive immune systems [32,33,34,35,36,37]. However, it has largely been demonstrated that MSCs are strongly influenced by the microenvironment of the so-called “stem cell niche”, which is able to drive the physiological MSCs phenotype towards an inflammatory profile, making them a source of pro-inflammatory cytokines capable of amplifying the inflammation according to a vicious circle [13].

In 2017, Orciani et al. [13] demonstrated that MSCs derived from skin samples of adult patients with chronic AD contribute to disease pathogenesis.

The results of the present study support our previous data, confirming the overexpression of IL6 and IL13 in AD-MSCs compared to C-MSCs. As extensively described in the literature, the lesional skin levels of IL13 and IL6 directly correlate with AD severity in adults [38,39,40,41,42].

In our case series, this trend is also confirmed in MSCs isolated from inflamed skin of adult AD patients, suggesting that the role of IL6 and IL13 can be backdated to MSCs.

The growing evidence of the involvement of IL4 and IL13 in the onset and development of AD has suggested that MSCs may be considered targets for molecular therapy.

The concept that AD may be an IL4-driven disorder emerges from the evidence that it is crucial in the regulation of the IgE synthesis, and several reports have emphasized the high response to and production of IL4 by appropriately activated lymphocytes isolated from AD lesional skin and in vivo overstimulation of the IL4/IL4R pathway [43]. However, despite this common perception that AD is an IL4-driven disease, data have consistently shown that while the expression of IL13 is always detected at high levels regardless of the methodology used, the level of expression of IL4 is changing, from high to low up to undetectable and strictly related to the experimental approach used [43]. These observations may explain our results: the expression of IL4 has been found to be faint, whereas the expression of IL13 was highly detected in all samples. According to our results, it has already been demonstrated that in subacute and chronic AD, IL13 mRNA is more expressed than IL4 mRNA [44,45]. The same trend observed for gene expression was also found at protein level in lesional skin, confirming high levels of IL13 in all skin samples, whereas IL4 expression was low or nondetectable in most patients [46]. Taken together, all these results focus on the idea that IL13 plays a central role in the skin manifestation related to AD, whereas IL4 drives the Th2 response, i.e., for activation in the lymph nodes [47].

All enrolled patients have been treated with Dupilumab for 16 weeks; all of them were good responders, reaching EASI75 (Table 1), and all clinometric (EASI, PGA, and SCORAD) significantly decreased over time (Table 2). Dupilumab is a totally human monoclonal antibody of the IgG4 subclass, a competitive antagonist of the α subunit of IL4Rα, shared by both the IL4 and IL13 receptors, which therefore inhibits intracellular signaling of both interleukins. Until the 2000s, the IL4 was considered the key player of AD, and the choice of Dupilumab in the treatment of AD was initially driven by its ability to block IL4. Nowadays, considering the swinging expression of IL4 compared with the strong and constant IL13 in AD, it is reasonable to suggest that the real target of Dupilumab in AD treatment is IL13. This hypothesis is supported by our results, which indicate that after treatment, both secretion and expression of IL13 (and IL6) by MSCs were normalized, whereas no significant changes have been detected for the other investigated cytokines. 

The dual inhibition of IL4 and IL13 has proven to be effective and with a synergic result, but taken individually, IL13 is certainly the major target in treatment strategy for AD. The pathogenetic role of IL-6 is also of interest; like IL-13, it is higher in AD-MSCs than in C-MSCs, and is reduced after Dupilumab administration.

It has already been demonstrated that Dupilumab Suppresses the Activation of Th2 and Th17/Th22, but its action on Th1 Immune Pathways is still unclear. Th1 cells release predominantly IL2, INF-gamma, and IL6 and Th2 cells release IL4, IL5, not INF-gamma. The increased IL6 production by atopic T cells may also result from the activation of a Th2 sub-set, which may represent the target of Dupilumab. However, it is also possible that Dupilumab also acts indirectly on the atopic Th1 subset, which our study results suggest, thereby reducing IL-6 production, but this will need to be investigated in further studies looking at cellular and molecular targets of the drug [48,49].

IL-6 is a pro-inflammatory cytokine produced by macrophages, dendritic cells (DC), and B cells that stimulates the acute-phase response, B-cell maturation, and macrophage differentiation [50]. In AD, IL-6 promotes Th2 differentiation, simultaneously inhibits Th1 polarization, and is involved in the transition from acute to chronic AD [51,52]. 

The inhibitory effect of Dupilumab on the expression of IL13 and IL6 in MSCs is of particular interest, since IL-6 and IL-13 are key molecules in the pathogenesis of atopic dermatitis, with different levels of involvement according to the clinical phase of disease, the former in the acute, and the latter in the chronic phase.

The evidence of IL6 and IL13 inhibition at the mesenchymal level configures an upstream therapeutic intervention, able to potentially modify the natural history of the disease, both in the acute and chronic phase of disease.

The value of IL-6 as therapeutic target in several inflammatory and immune mediated has been postulated on psoriasis [53] and on atopic dermatitis, as demonstrated by data on Tocilizumab, a humanized monoclonal antibody directed against the IL-6 receptor, which in three case series demonstrated efficacy on pruritus and EASI index, in patients refractory to topical corticosteroid therapy and, in two cases, to cyclosporine [54].

The role of IL6 in psoriasis comorbidities (e.g., depression) has been established [55,56], whereas its involvement in atopic dermatitis comorbidities is far from being proven, although data from literature seem to indicate that the prevalence of depression among atopic patients is higher than in the general population [57], and the role of IL6 need to be investigated in more detail. In this regard, in 2020, He et al. emphasized the potential utility of tape-strip proteomic profiling for tracking biomarkers of therapeutic response in real-life settings, as well as clinical trials and longitudinal studies of AD [48].

However, several emerging items need to be clarified through further studies. It is unclear which pathway dupilumab takes in exerting inhibitory action on IL6 expression, and comprehension of this method of action could have pathogenetic and perhaps therapeutic relevance. Moreover, given the clinical efficacy of Tocilizumab, it might be interesting to evaluate modification of MSCs immunophenotype under the effect of an IL-6 inhibitor. In conclusion, our studies highlight the efficacy of Dupilumab at clinical and subclinical level, focusing on its effect on AD-MSCs immunophenotypic profile.

Further studies focusing on the changes in inflammatory immunophenotype of MSCs obtained from nonresponding patients could be of interest. However, in accordance with the protocol approved by our local ethics committee, we were not allowed to perform skin biopsies to evaluate skin changes in nonresponding patients.

An interesting implication of our results could also be to investigate whether the effect of target molecular therapy on MSCs might be able to restore the typical anti-inflammatory profile of naive MSCs. This could clarify the therapeutic potential of regenerative medicine with the use of autologous MSCs in treating AD [58].

To the best of our knowledge, only one clinical trial (phase I/IIa) has been conducted on human-umbilical-cord-derived mesenchymal stem cells (hUCB-MSCs) to treat adult patients with AD. This trial has demonstrated the potential efficacy of hUCB-MSCs with 50% reduction in EASI in 6 out of 11 subjects, with no reported side effects [59]. Therefore, in future therapeutic strategies for AD, both an increase in knowledge of AD-MSCs and the mechanisms of action exerted on them by the target molecular therapies are crucial to integrate the current pharmacological approach with regenerative medicine, whose preliminary results look promising.

## Figures and Tables

**Figure 1 jcm-11-04759-f001:**
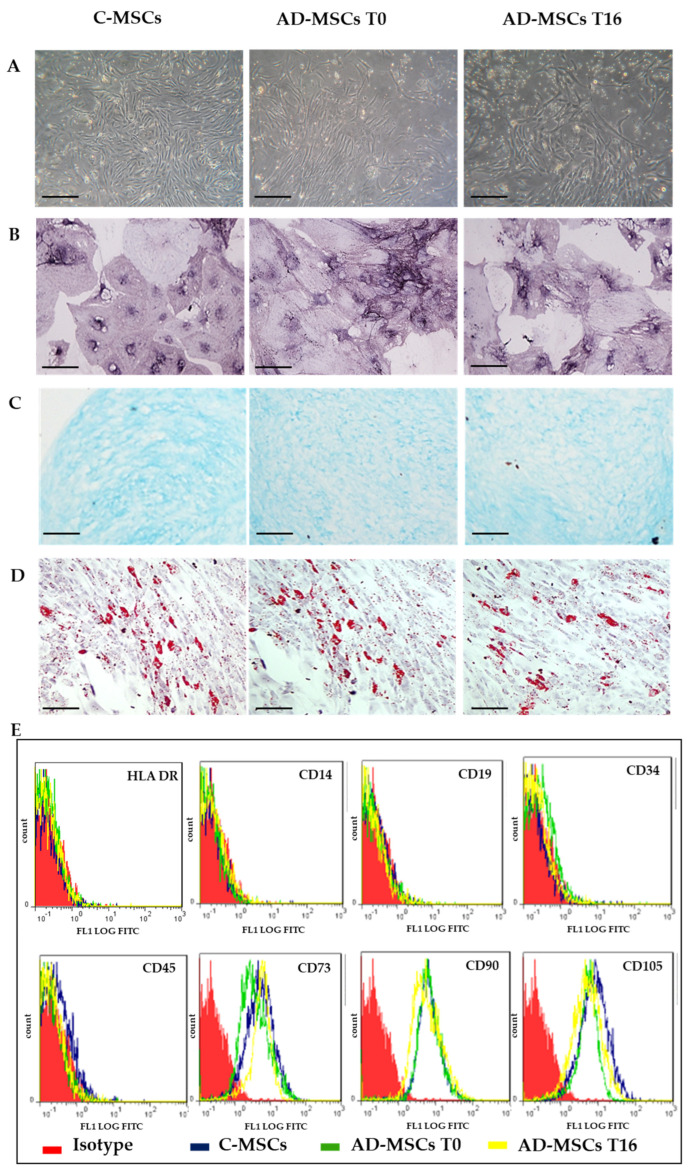
Representative images for Mesenchymal Stem Cells (MSCs) characterization. (**A**) Phase-contrast images of MSCs derived from skin of healthy control subject (C-MSC) and of patients with atopic dermatitis (AD-MSC) before (T0) and after (T16) treatment with Dupilumab. Scale bar: 100 µm; (**B**) Osteogenic differentiation, ALP staining, Scale bar: 50 µm; (**C**) Chondrogenic differentiation, Alcian blue staining, Scale bar: 200 µm; (**D**) Adipogenic differentiation, Oil red staining, Scale bar: 200 µm; (**E**) Flow cytometry analyses of cell-surface antigen expression, as indicated. Red histograms refer to the negative control (IgG1 isotype control–fluorescein isothiocyanate (FITC) labeled). Blue histograms: C-MSCs; Green: AD-MSCs T0; Yellow: AD-MSCs T16.

**Figure 2 jcm-11-04759-f002:**
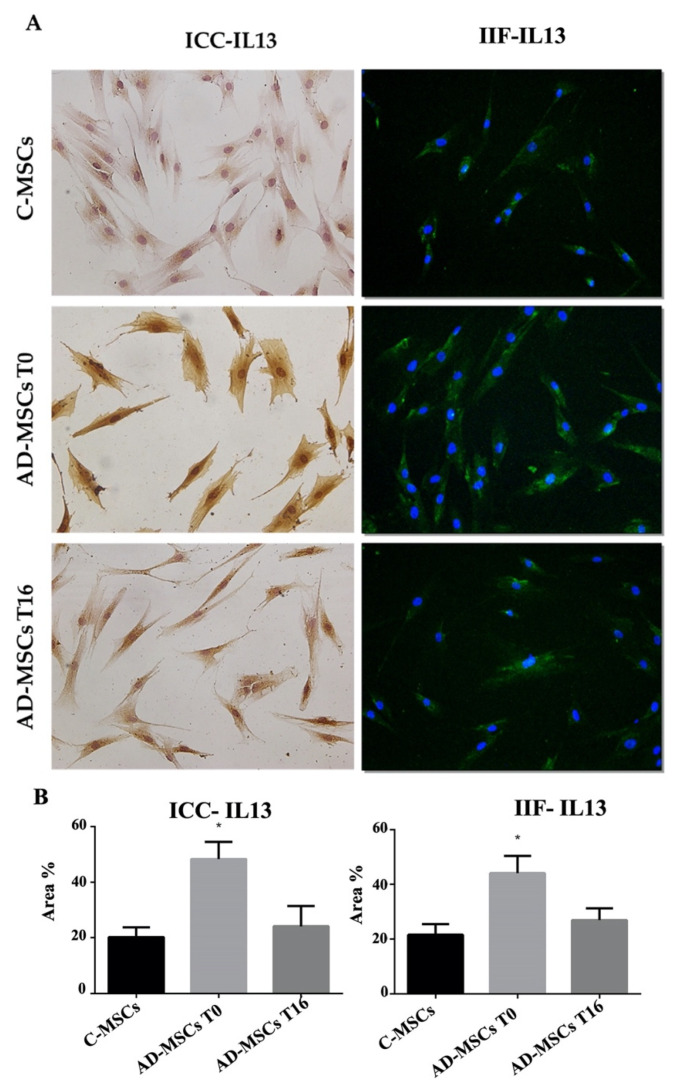
Analysis of the expression of IL13 by Immuno-cytochemistry (ICC), and indirect immune-fluorescence (IIF). (**A**) Representative images of IIF and ICC of IL13 on MSCs from healthy controls (C-MSCs), and MSCs from patients with AD (AD-MSCs) at T0 and at T16. For IIF, a secondary FITC-conjugated antibody was used after incubation with the primary antibodies. Nuclei were counterstained with Hoechst 33342. For ICC, slides were treated with 3,3-diaminobenzidine and counterstained with Mayer’s hematoxylin. (Scale bar: 100 µm). (**B**) Quantification of proteins expression processed by Fiji-ImageJ. Protein expression is represented by the percentage of the area it occupies inside the cell. The * indicates significative differences of C-MSCs vs. AD-MSC (unpaired *t*-test; *p* < 0.05).

**Figure 3 jcm-11-04759-f003:**
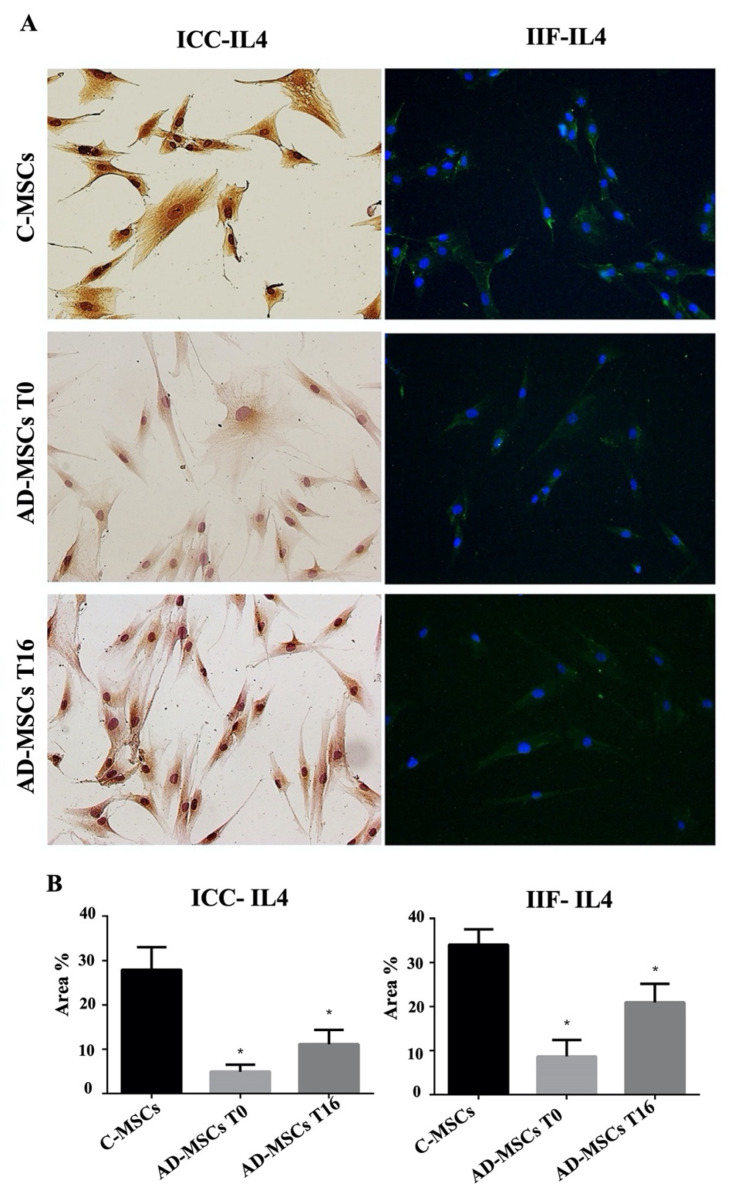
Analysis of the expression of IL4 by Immuno-cytochemistry (ICC) and indirect immune-fluorescence (IIF). (**A**) Representative images of IIF and ICC of IL4 on MSCs from healthy controls (C-MSCs), and MSCs from patients with AD (AD-MSCs )at T0 and at T16. For IIF, a secondary FITC-conjugated antibody was used after incubation with the primary antibodies. Nuclei were counterstained with Hoechst 33342. For ICC, slides were treated with 3,3-diaminobenzidine and counterstained with Mayer’s hematoxylin. (Scale bar: 100 µm). (**B**) Quantification of proteins expression processed by Fiji-ImageJ. Protein expression is represented by the percentage of the area it occupies inside the cell. The * indicates significative differences of C-MSCs vs. AD-MSC (unpaired *t*-test; *p* < 0.05).

**Figure 4 jcm-11-04759-f004:**
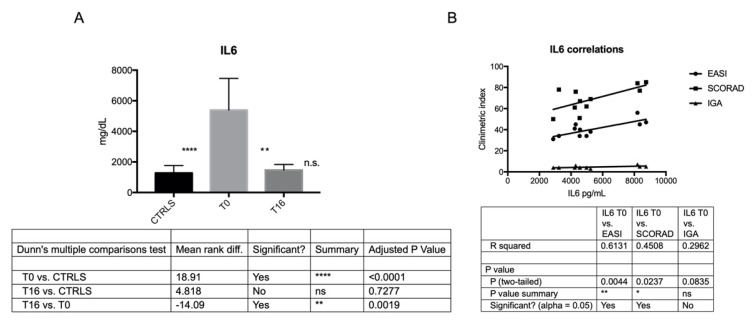
IL6 secretion by C- and AD-MSCs at T0 and T16 and its correlation with AD severity at baseline. (**A**) The histogram depicts the level of secreted IL6 in C-MSCs and in AD-MSCs at T0 and at T16. Secreted protein, evaluated in triplicates by ELISA test, has been reported as pg/mL. ****: *p* < 0.0001, AD-MSCs T0 vs. C-MSCs; **: *p* = 0.0019, AD-MSCs T0 vs. AD-MSCs T16. (**B**) Correlation between IL6 secreted by MSCs at baseline and AD severity according to EASI, SCORAD, and IGA. ** *p* = 0.0044; * *p* = 0.0237.

**Figure 5 jcm-11-04759-f005:**
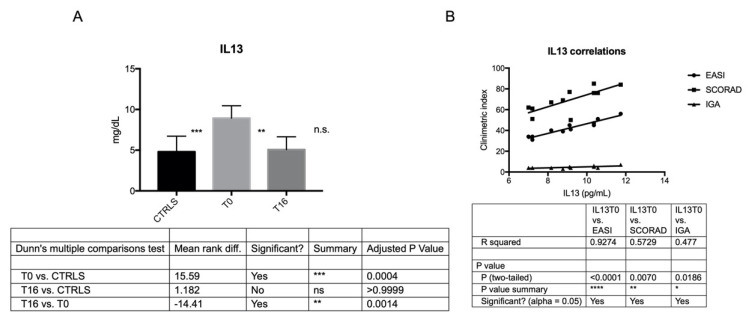
IL13 secretion by C-and AD-MSCs at T0 and T16 and its correlation with AD severity at baseline. (**A**) The histogram depicts the level of secreted IL13 in C-MSCs and in AD-MSCs at T0 and at T16. Secreted protein, evaluated in triplicates by ELISA test, has been reported as pg/mL. ***: *p* < = 0.0004, AD-MSCs T0 vs. C-MSCs; **: *p* = 0.0014, AD-MSCs T0 vs. AD-MSCs T16. (**B**) Correlation between IL13 secreted by MSCs at baseline and AD severity according to EASI, SCORAD, and IGA. **** *p* < 0.0001; ** *p* = 0.0070; * *p* = 0.0186. Conversely, IL4 was secreted at lower levels by AD-MSCs than C-MSCs both at T0 and T16 (Table 3). Finally, IL2, IL5, IL10, IL12, IL17A, TNFα, IFNgamma, G-CSF, TGFβ1 were similarly secreted by C- and AD-MSCs (Table 3).

**Table 1 jcm-11-04759-t001:** Absolute values of clinometric indexes in AD patients at baseline (T0) and after 16 weeks of treatment with Dupilumab (T16). (EASI: “Eczema Area and Severity Index”; PGA: “Physician Global Assessment”; SCORAD: “SCORing Atopic Dermatitis”).

Patients (*n* = 11)	EASI T0	SCORAD T0	PGA T0	EASI T16	SCORAD T16	PGA T16
1	45	76	6	11	20	2
2	56	84	6	13	22	2
3	34	62	4	8	15	2
4	33	51	4	7	13	2
5	45	77	5	11	20	2
6	40	67	4	10	17	2
7	41	61	4	10	16	2
8	47	85	5	11	21	2
9	31	50	4	7	12	2
10	34	78	4	9	23	3
11	38	69	3	11	26	2

**Table 2 jcm-11-04759-t002:** Clinometric changes after 16 weeks of treatment with Dupilumab. (EASI: “Eczema Area and Severity Index”; IGA: “Investigator’s Global Assessmen”; SCORAD: “SCORing Atopic Dermatitis”).

**EASI T0 Mean Value ± SD**	**EASI T16 Mean Value ± SD**	*p* < 0.0001
41.79 ± 2.38	9.81 ± 1.8	
**SCORAD T0 Mean Value ± SD**	**SCORAD T16 Mean Value ± SD**	*p* < 0.0001
69.09 ± 11.15	18.64 ± 1.32	
**IGA T0 Mean Value ± SD**	**IGA T16 Mean Value ± SD**	*p* < 0.0001
4.45 ± 0.93	2.09 ± 0.30	

**Table 3 jcm-11-04759-t003:** Th1, Th2, Th17 cytokines expression by ELISA in MSCs from healthy controls (C-MSCs) and MSCs from AD patients (AD-MSCs) at T0 and T16. C-MSCs.

IL	MSCs-AD T0 Mean Value ± SD	MSCs-AD T16 Mean Value ± SD	C-MSCs Mean Value ± SD	*p*
**IL2**	48.98 ± 6.73	65.75 ± 13.83	35.56 ± 8.54	T0 vs. Controls *p* = 0.0523
				T0 vs. T16 *p* = 0.1299
**IL4**	7.14 ± 0.89	9.53 ± 1.31	20.71 ± 1.92	T0 vs. Controls *p* < 0.0001
				T0 vs. T16 *p* = 0.0777
**IL5**	4.40 ± 0.81	3.65 ± 0.55	3.64 ± 0.38	T0 vs. Controls *p* = 0.1088
				T0 vs. T16 *p* = 0.1213
**IL6**	5387.03 ± 2073	1469.01 ± 364.9	1277.02 ± 483.1	T0 vs. Controls *p* < 0.0001
				T0 vs. T16 *p* = 0.0019
**IL10**	29.80 ± 5.82	31.55 ± 6.45	34.66 ± 6.11	T0 vs. Controls *p* = 0.1399
				T0 vs. T16 *p* > 0.9999
**IL12**	3.32 ± 3.12	6.17 ± 3.48	26.38 ± 3.48	T0 vs. Controls *p* = 0.1399
				T0 vs. T16 *p* > 0.9999
**IL13**	8.937 ± 1.51	5.0 ± 1.5	4.807 ± 1.91	T0 vs. Controls *p* = 0.0004
				T0 vs. T16 *p* = 0.0014
**IL17A**	5.90 ± 0.86	6.13 ± 1.19	5.31 ± 0.89	T0 vs. Controls *p* = 0.3628
				T0 vs. T16 *p* > 0.9999
**IFNγ**	6.19 ± 0.92	5.13 ± 0.88	4.81 ± 1.44	T0 vs. Controls *p* = 0.1170
				T0 vs. T16 *p* = 0.0808
**TNFα**	81.46 ± 19.48	66.60 ± 6.55	62.89 ± 11.52	T0 vs. Controls *p* = 0.0639
				T0 vs. T16 *p* = 0.4878
**G-CSF**	94.07 ± 80.53	68.53 ± 38.44	30.93 ± 3.25	T0 vs. Controls *p* = 0.6532
				T0 vs. T16 *p* > 0.9999
**TGFβ**	843.09 ± 170.40	641.10 ± 155.70	759.5 ± 302.70	T0 vs. Controls *p* = 0.0807
				T0 vs. T16 *p* > 0.9999

Cytokines’ concentration is expressed in pg/mL. Experiments were performed in triplicates.

## Data Availability

The data presented in this study are available on request from the corresponding author.

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
