# Peer review of "Mesenchymal Stem Cells Profile in Adult Atopic Dermatitis and Effect of IL4-IL13 Inflammatory Pathway Inhibition In Vivo: Prospective Case-Control Study"

_jcm, 2022, doi:10.3390/jcm11164759_

Round 1

Reviewer 1 Report

Dear Authors,

I thank you for your efforts to provide a well-written manuscript with solid data on the effects of dupilumab on the MSCs of AD patients.

I would recommend the following points for the improvement of your manuscript:

You tend to write single sentences. Please merge the sentence into paragraphs. That makes the manuscript easier to the readers to follow.

Methods and Materials:

Line 82: avoid the term “intrinsic”

Line 88: change “phlogistic” to “inflammatory”

Results:

Figure 1 can be omitted. It does not add any information to the manuscript.

Figure 2B: “AD-MSCs-T12” should be corrected to “AD-MSCs-T16”

Figure 3B: as mentioned in Figure2B

Figure 4B: Authors should separately show the correlation of IL-6 with SCORAD, EASI and IGA. Especially in the case of IGA, due to the scaling of y-axis the correlation can be clearly presented.

Figure 5B: as mentioned in Figure 4B

Text corrections:

Line 292: “activation of the lymph nodes” should be changed to “activation in the lymph nodes”

Line 322: delete “clinometric”

Line 324: replace “it” with “It”

Line 338: replace “recorded” with “reported”

You should discuss, what is the underlying mechanism of the effects you show. Do MSCs express IL-4R or IL-13R? Is this downstream known?

The conclusion that the reduction of IL-6 is a specific effect of dupilumab is not supported by the setting of these experiments. A control group of another therapy (ciclosporin? UV? JAKinhibitor?) is needed for that.

I thank you for considering my comments.

Kind regards

Author Response

You tend to write single sentences. Please merge the sentence into paragraphs. That makes the manuscript easier to the readers to follow. DONE

Methods and Materials:

Line 82: avoid the term “intrinsic” DONE

Line 88: change “phlogistic” to “inflammatory” DONE

Results:

Figure 1 can be omitted. It does not add any information to the manuscript. Although we agree about the opportunity to remove fig. 1, we think that image is exciting, this we prefer to maintain it in the revised version, if we could do it.

Figure 2B: “AD-MSCs-T12” should be corrected to “AD-MSCs-T16” DONE

Figure 3B: as mentioned in Figure2B DONE

Figure 4B: Authors should separately show the correlation of IL-6 with SCORAD, EASI and IGA. Especially in the case of IGA, due to the scaling of y-axis the correlation can be clearly presented. As suggested by reviewer correlation of IL-6 and clinicometric indices are clear presented within the graph.  

Figure 5B: as mentioned in Figure 4B As above.

Text corrections:

Line 292: “activation of the lymph nodes” should be changed to “activation in the lymph nodes” DONE

Line 322: delete “clinometric” DONE

Line 324: replace “it” with “It” DONE

Line 338: replace “recorded” with “reported” DONE

You should discuss, what is the underlying mechanism of the effects you show. Do MSCs express IL-4R or IL-13R? Is this downstream known? The molecular target of Dupilumab is the alpha subunit of IL-4 receptor, so it is likely that MSCs express, like cells involved in the Th2 response, this receptor. Our study demonstrates a decrease in IL-13 secretion by atopic MSCs after Dupilumab administration, suggesting a key role of this molecule over IL-4 in disease pathogenesis. Studies inherent in the receptors expressed by atopic MSCs are lacking and could be conducted in furtherance of our work.  However the aim of this study, beyond the specific expression of IL-4R and Il13-R in  MSCs, is to focus on the effect of stem cell niche immunomodulation on MSCs immunophenotypic profile.

The conclusion that the reduction of IL-6 is a specific effect of dupilumab is not supported by the setting of these experiments. A control group of another therapy (ciclosporin? UV? JAKinhibitor?) is needed for that. We believe that the role of IL-6 in the pathogenesis of atopic dermatitis is yet to be defined, both with respect to the Th1 response and the Th2 response. The molecular mechanism by which Dupilumab would induce in MSCs a reduction in secretion of this molecule needs to be elucidated with further studies, both molecular and clinical. In this regard, studies with control groups of another therapy can certainly be useful.

Reviewer 2 Report

The authors present interesting data on the role of dupilumab treatment and its effect on MSCs in patients with atopic dermatitis. Studies on the modulation of MScs after treatment are limited in the literature .

The role of IL-6 in AD is of particular interest. The authors could comment on the recently reported findings that increased levels of IL-6 have been associated with depression and a depressive dimension of affective temperament. in patients with psoriasis. (Luiza Marek-Józefowicz et al. Postepy Dermatol Alergol. 2021) Patients with AD often present with depressive symptoms. A possible link with IL-6 is a promising field to explore.

The role of IL-6 as a potential biomarker could also be briefly discussed (He et al Front Immunol 2020)

Thank you.

Author Response

The authors present interesting data on the role of dupilumab treatment and its effect on MSCs in patients with atopic dermatitis. Studies on the modulation of MScs after treatment are limited in the literature.

 The role of IL-6 in AD is of particular interest. The authors could comment on the recently reported findings that increased levels of IL-6 have been associated with depression and a depressive dimension of affective temperament. in patients with psoriasis. (Luiza Marek-Józefowicz et al. Postepy Dermatol Alergol. 2021) Patients with AD often present with depressive symptoms. A possible link with IL-6 is a promising field to explore.

Comments on the role of IL6 in psoriasis and possible link with atopic dermatitis has been included following the suggestion of reviewer.

 The role of IL-6 as a potential biomarker could also be briefly discussed (He et al Front Immunol 2020). Role of IL6 as potential biomarker has been discussed, following the reviewer suggestion.

Reviewer 3 Report

Comment to Authors

The manuscript entitled as “Mesenchimal stem cells profile in adult atopic dermatitis and effect of IL4-IL13 inflammatory pathway inhibition in vivo: prospective case-control study” revealed that Dupilumab, antibody against IL4 receptor α, reduced the expression of IL13 and IL6 in atopic dermatitis patient-derived mesenchymal stem cells (AD-MSCs). The authors conclude that Dupilumab inhibits IL4 and IL13 signaling, resulting reduction of Th2 response. However, there are several shortfalls in this manuscript. Authors need to carefully explain how IL4 receptor inhibitor, Dupilumab, reduced the levels of IL13 and IL6 in AD-MSCs. 

Major

1. Author explained that Dupilumab would specifically modulate the expression of IL4 and IL13 (lane 192). Where are these results shown? Moreover, which cells do authors think the target of Dupilumab? 

Why do authors think IL13 and IL6 expressions were reduced by inhibiting the IL4 and IL13 signal pathway? Authors have not discussed the above.   

2. This study does not included the AD patients who did not have therapeutic effect. All AD- MSCs were established from the skin of AD patients who were good responders of Dupilumab. Was there a change of IL13 and IL6 expressions after Dupilumab administration in MSCs of AD patients who did not respond to Dupilumab? The causal relationship between changes in cytokine expression and the therapeutic effect of Dupilumab need to be clarified.

3. Authors focus IL4 and IL13, as the Th2 immune response, and IL6, as Th1 immune response. In introduction (lane 73-75), it was written that the relative expression of selected Th1, Th2, and Th17 chemokines/cytokines has been analyzed in MSCs obtained from healthy subjects. However, the reason for examining Th1 and Th17 cytokine was not explained. Authors must explain this point.

4. What was the purity of the MSCs used in this study? And how was it verified that these were authentic MSCs? The data must be show.

5. Also, of interest appears to be the pathogenetic role of IL-6, which like IL-13 is higher 307 in AD-MSCs than in C-MSCs and is reduced after Dupilumab administration (reference 45) (lanes 307-308). The above data were shown in this study?

Minor

1. Abbreviations not used correctly. For example, IL4 in lane20, MSCs in lane 21, assay in lane 28, cytokines in lane 28 in abstract and cytokines in lane 56 in the text. And atopic dermatitis (lane 104) must be AD 

2. IL4 receptor α subunit was shown IL4α (lane 72). Is this expression common?

3. What mean SC (lane 114)?

4. In Figure 2A, the three photos were not take under the same conditions.

5. In 2017 Orciani et al.13 (lane 266) must be corrected.

Author Response

The manuscript entitled as “Mesenchimal stem cells profile in adult atopic dermatitis and effect of IL4-IL13 inflammatory pathway inhibition in vivo: prospective case-control study” revealed that Dupilumab, antibody against IL4 receptor α, reduced the expression of IL13 and IL6 in atopic dermatitis patient-derived mesenchymal stem cells (AD-MSCs). The authors conclude that Dupilumab inhibits IL4 and IL13 signaling, resulting reduction of Th2 response. However, there are several shortfalls in this manuscript. Authors need to carefully explain how IL4 receptor inhibitor, Dupilumab, reduced the levels of IL13 and IL6 in AD-MSCs.

Many thanks for your suggestion, Dupilumab has entered the therapeutic armamentarium of AD in the recent years, owing to its selective action on IL4-IL13, by blocking the IL-4R alpha subunit, dupilumab inhibits IL-4 and IL-13 cytokine-induced responses, including the release of proinflammatory cytokines, chemokines, and immunoglobulin E.

Major

  1. Author explained that Dupilumab would specifically modulate the expression of IL4 and IL13 (lane 192). Where are these results shown? Moreover, which cells do authors think the target of Dupilumab? Starting form evidence on literature demonstrating the ability of  dupilumab to modulate the expression of IL4 and IL13 in differentiated immune cells [Hamilton JD, Suárez-Fariñas M, Dhingra N, Cardinale I, Li X, Kostic A, Ming JE, Radin AR, Krueger JG, Graham N, Yancopoulos GD, Pirozzi G, Guttman-Yassky E. Dupilumab improves the molecular signature in skin of patients with moderate-to-severe atopic dermatitis. J Allergy Clin Immunol. 2014 Dec;134(6):1293-1300. Doi: 10.1016/j.jaci.2014.10.013. PMID: 25482871.; Ariëns LFM, Bakker DS, van der Schaft J, Garritsen FM, Thijs JL, de Bruin-Weller MS. Dupilumab in atopic dermatitis: rationale, latest evidence and place in therapy. Ther Adv Chronic Dis. 2018 May 11;9(9):159-170. Doi: 10.1177/2040622318773686. PMID: 30181845; PMCID: PMC6116085.], as no data have been already published on its effects on MSCs, the aim of this study was to evaluate  the effect of Dupilumab on MSCs, whose immunomodulatory action is well known.

Why do authors think IL13 and IL6 expressions were reduced by inhibiting the IL4 and IL13 signal pathway? Authors have not discussed the above. 

It has already been demonstrated that Dupilumab is able to suppress the activation of Th2 and Th17/Th22, but its action on Th1 Immune Pathways is still unclear. Th1 cells release predominantly IL2, INF-gamma, and IL6 and Th2 cells release IL4, IL5, not INF-gamma. However, the increased IL6 production by atopic T cells may result also from the activation of a Th2 sub-set, which may represent the target of dupilumab. It is also possible that Dupilumab acts indirectly on the atopic Th1 subset, which our study results suggest, thereby reducing IL-6 production, but this will need to be investigated in further studies looking at cellular and molecular targets of the drug. [He H, Olesen CM, Pavel AB, Clausen ML, Wu J, Estrada Y, Zhang N, Agner T, Guttman-Yassky E. Tape-Strip Proteomic Profiling of Atopic Dermatitis on Dupilumab Identifies Minimally Invasive Biomarkers. Front Immunol. 2020 Aug 6;11:1768. doi: 10.3389/fimmu.2020.01768. PMID; Toshitani A, Ansel JC, Chan SC, Li SH, Hanifin JM. Increased interleukin 6 production by T cells derived from patients with atopic dermatitis. J Invest Dermatol. 1993 Mar;100(3):299-304. doi: 10.1111/1523-1747.ep12469875. PMID: 8440909.].

  1. This study does not included the AD patients who did not have therapeutic effect. All AD- MSCs were established from the skin of AD patients who were good responders of Dupilumab. Was there a change of IL13 and IL6 expressions after Dupilumab administration in MSCs of AD patients who did not respond to Dupilumab? The causal relationship between changes in cytokine expression and the therapeutic effect of Dupilumab need to be clarified.

 Our results aim to evaluate the effect of dupilumab on MSCs profile, rather than to establish a strict causal relationship between MSCs changes and clinical effect of dupilumab. Moreover, although the opportunity to make a comparison between responding and non-responding patients it is a very intriguing matter it was not possible to make it, as in accordance with the protocol approved by our local ethics committee, skin biopsies were not allowed to evaluate skin changes in non-responding patients. For this reason, we report that performing further studies focusing on the changes in inflammatory immunophenotype of MSCs obtained from non responding patients could be of interest, and this has been reported among the limits of the study in the revised version of the manuscript

  1. Authors focus IL4 and IL13, as the Th2 immune response, and IL6, as Th1 immune response. In introduction (lane 73-75), it was written that the relative expression of selected Th1, Th2, and Th17 chemokines/cytokines has been analyzed in MSCs obtained from healthy subjects. However, the reason for examining Th1 and Th17 cytokine was not explained. Authors must explain this point.         

Recent studies on topical dermatitis, performed, for example, on skin biopsies or peripheral blood, have shown that the Th2 and Th17 response are characteristic of the acute phase of disease, while the Treg and Th1 response play a major role in chronic phase. Thus, it was our intention to evaluate the expression of these molecules also in MSCs from patients with atopic dermatitis, and compare it with that of MSCs from healthy patients, to delineate the immunophenotype of atopic stem cells in the different stages of disease. Su, C.; Yang, T.; Wu, Z.; Zhong, J.; Huang, Y.; Huang, T.; Zheng, E. Differentiation of T-helper cells in distinct phases of atopic dermatitis involves Th1/Th2 and Th17/Treg. Eur. J. Inflamm. 2017, 15, 46–52.

  1. What was the purity of the MSCs used in this study? And how was it verified that these were authentic MSCs? The data must be show.

As reported in the text, we followed the criteria for mesenchymal cells identification accepted by the International Society for Cellular Therapy (Dominici M et al. Minimal criteria for defining multipotent mesenchymal stromal cells. The International Society for Cellular Therapy position statement. Cytotherapy. 2006;8(4):315-7. doi: 10.1080/14653240600855905).

In detail, 3 conditions must be addressed: 1) plastic adherence in culture; 2) immunophenotype characterized by expression of CD73, CD90, CD105 and no expression of HLA-DR, CD14, CD19, CD34, CD45; 3) differentiative potential towards osteogenic, chondrogenic and adipogenic lineages.

More than 90% of isolated cells satisfied these criteria and, according to your suggestions, we have included these results in the text.

  1. Also, of interest appears to be the pathogenetic role of IL-6, which like IL-13 is higher 307 in AD-MSCs than in C-MSCs and is reduced after Dupilumab administration (reference 45) (lanes 307-308). The above data were shown in this study?

We have explicated the data on this in results section paragraph 3.4 Expression profiles of Th1, Th2, Th17 cytokines by ELISA in C-MSCs and AD-MSCs at T0 and T16, with also the graph displayed in Fig.4

 Minor

  1. Abbreviations not used correctly. For example, IL4 in lane20, MSCs in lane 21, assay in lane 28, cytokines in lane 28 in abstract and cytokines in lane 56 in the text. And atopic dermatitis (lane 104) must be AD

The use of abbreviations has been revised throughout the text

  1. IL4 receptor α subunit was shown IL4α (lane 72). Is this expression common?

The nomenclature has been corrected in the text.

  1. What mean SC (lane 114)? It means “subcutaneous”. The nomenclature has been corrected in the text.

  1. In Figure 2A, the three photos were not take under the same conditions.

Figure 2A displays representative images derived from immunocytochemistry (ICC) and immunofluorescence (IIF) analyses of IL13. Each sample was tested in duplicate for IIC and IIF and, for each reaction, a quantitative analysis was performed by calculating the percentage of the positive area of 10 cells for 3 different fields.

According to the results of the statistical analysis, representative images were chosen even if from different experiments.

As you suggested we replaced them with images acquired from the same experiment.

  1. In 2017 Orciani et al.13 (lane 266) must be corrected.

Reference has been corrected.

Round 2

Reviewer 3 Report

Comment to Authors

1. Authors responded to questions from reviewer, but the responses were not satisfactory enough.

In this study, Mesenchimal stem cells (MSCs) were prepared from atopic dermatitis (AD) patients who treated with dupliumab, IL-4 receptor α inhibitor, and then the expression of cytokines in cells were analyzed. Duplimab may have acted directly and /or indirectly on MSCs. Thus, the increase in the IL4 expression and the decrease in the IL13and IL6 expressions in MSCs are the result of the drug acting on IL4 receptors in various cells in the body.

Authors responded that aim of this study was to evaluate the effect of Dupilumab on MSCs, whose immunomodulatory action is well known. However, this is a study analyzed cytokine productions by MSCs after treatment of Duplimab in AD patients. If authors want to show the effect of Duplimab on MSCs, Duplimab should be used on MSCs prepared from AD patients before treatment of Duplimab. Or better yet, analyze the course of treatment and the case in which Duplimab did not respond to treatment.

2. In abstract, IL4α is a mistake.

3. The revised Figure 1, it is shown AD-MSCs T12 at the top of the photo. Authors need to show the data of AD-MSCs T16.

4. It needs the data of histogram or dot plot in Flow cytometry 

5. The bar graph quantifying the photo do not seem to match the photos in Figures 2 and 3. Quantitative results especially for IIF.

6. As reviewer pointed out in your first review, lanes 435-437(first manuscript; lane 307-308) is the data in this study. Why was reference (48) cited?

Author Response

Reviewer#3

  1. Authors responded to questions from reviewer, but the responses were not satisfactory enough.

In this study, Mesenchimal stem cells (MSCs) were prepared from atopic dermatitis (AD) patients who treated with dupliumab, IL-4 receptor α inhibitor, and then the expression of cytokines in cells were analyzed. Duplimab may have acted directly and /or indirectly on MSCs. Thus, the increase in the IL4 expression and the decrease in the IL13and IL6 expressions in MSCs are the result of the drug acting on IL4 receptors in various cells in the body.

Authors responded that aim of this study was to evaluate the effect of Dupilumab on MSCs, whose immunomodulatory action is well known. However, this is a study analyzed cytokine productions by MSCs after treatment of Duplimab in AD patients. If authors want to show the effect of Duplimab on MSCs, Duplimab should be used on MSCs prepared from AD patients before treatment of Duplimab. Or better yet, analyze the course of treatment and the case in which Duplimab did not respond to treatment.

We thank you for your clarifications, which provide us with the opportunity to go into detail in our explanation. Our aim was to evaluate changes of the in vivo immunohistochemical profile of MSCs in patients with atopic dermatitis who underwent treatment with dupilumab and benefited from this type of therapy.

As you have suggested, we isolated MSCs from skin of patients both before (AD-MSCs T0) and after (AD-MSCs T16) dupilumab administration, to compare their profile; in addition,  we compared AD-MSCs (both T0 and T16) with MSCs derived from healthy control subjects (C-MSCs) with the aim of evaluating whether treatment with dupilumab was associated with a trend toward normalization of MSCs profile. This is in accordance with your suggestion to evaluate the course of treatment with Dupilumab, and this type of investigation has a design that has been described and widely accepted in the literature in many our other studies conducted on MSCs in inflammatory and immune-mediated diseases.

Illustrative and non-exhaustive list of publications with similar design, on relevant scientific journals are the following:

Campanati A, Caffarini M, Diotallevi F, Radi G, Lucarini G, Di Vincenzo M, Orciani M, Offidani A. The efficacy of in vivo administration of Apremilast on mesenchymal stem cells derived from psoriatic patients. Inflamm Res. 2021 Jan;70(1):79-87. doi: 10.1007/s00011-020-01412-3. Epub 2020 Nov 18. PMID: 33210178.

Campanati A, Bobyr I, Sorgentoni G, Diotallevi F, Caffarini M, Pellegrino P, Di Primio R, Offidani A, Orciani M. Mesenchymal stem cell profile in actinic keratosis and its modification after topical application of ingenol mebutate. J Eur Acad Dermatol Venereol. 2020 Mar;34(3):e148-e149. doi: 10.1111/jdv.16058. PMID: 31709665.

Campanati A, Orciani M, Lazzarini R, Ganzetti G, Consales V, Sorgentoni G, Di Primio R, Offidani A. TNF-α inhibitors reduce the pathological Th1 -Th17 /Th2 imbalance in cutaneous mesenchymal stem cells of psoriasis patients. Exp Dermatol. 2017 Apr;26(4):319-324. doi: 10.1111/exd.13139. Epub 2016 Oct 24. PMID: 27376466.

Campanati A, Orciani M, Ganzetti G, Consales V, Di Primio R, Offidani A. The effect of etanercept on vascular endothelial growth factor production by cutaneous mesenchymal stem cells from patients with psoriasis. J Int Med Res. 2016 Sep;44(1 suppl):6-9. doi: 10.1177/0300060515593229. PMID: 27683131; PMCID: PMC5536541.

Campanati A, Orciani M, Gorbi S, Regoli F, Di Primio R, Offidani A. Effect of biologic therapies targeting tumour necrosis factor-α on cutaneous mesenchymal stem cells in psoriasis. Br J Dermatol. 2012 Jul;167(1):68-76. doi: 10.1111/j.1365-2133.2012.10900.x. Epub 2012 Jun 6. PMID: 22356229.

Finally, even if  your suggestion to compare the MSCs features between unresponsive and responsive patients to Dupilumab is interesting, this evaluation deviates from our purpose (to evaluate the effects of Dupilumab on MSCs) and, in addition, it would require a different experimental approach.

in this regards, if after 16 weeks of treatment a patient would be recognized as NON RESPONSIVE, it may be not ethical (and not approved by our local Ethical committee) to repeat an invasive procedure like skin biopsy.

A hypothesis could be an in vitro administration with Dupilumab of MSCs-T0 derived from responsive and unresponsive patients, followed by the analysis of cytokines to evaluate why these patients are unresponsive.

In this line, your suggestion may be a valid idea for a next research work, aimed to understand the molecular differences between responsive and unresponsive patients, but using an in vitro (instead of an in vivo) study design.

  1. In abstract, IL4α is a mistake

Thanks, the typo has been corrected in IL4Rα.

  1. The revised Figure 1, it is shown AD-MSCs T12 at the top of the photo. Authors need to show the data of AD-MSCs T16.

We apologize for the typing mistake, and we changed with the correct sampling time T16

  1. It needs the data of histogram or dot plot in Flow cytometry 

Flow cytometry histograms have been added

  1. The bar graph quantifying the photo do not seem to match the photos in Figures 2 and 3. Quantitative results especially for IIF.

As we specified in the last round of revision, each sample was tested in duplicate for IIC and IIF and, for each reaction, a quantitative analysis was performed by calculating the percentage of the positive area of 10 cells for 3 different fields. Here we display just one representative picture, but the histograms are the results of the analyses of more photos and so it may be that it does not match perfectly

  1. As reviewer pointed out in your first review, lanes 435-437(first manuscript; lane 307-308) is the data in this study. Why was reference (48) cited?

Ref 48 has been reported as data from literature supporting at least in part our results. However, as you think it is superfluous, it has been removed.